# To Achieve a Bullseye: Factors Related to Corneal Refractive Therapy Orthokeratology Lens Toricity

**DOI:** 10.3390/jcm11195635

**Published:** 2022-09-24

**Authors:** Changfei Li, Li Zeng, Jiaqi Zhou, Bingjie Wang, Zhi Chen

**Affiliations:** 1Department of Ophthalmology, Qilu Hospital (Qingdao), Cheeloo College of Medicine, Shandong University, 758 Hefei Road, Qingdao 266035, China; 2Department of Ophthalmology and Vision Science, Eye and ENT Hospital, Fudan University, Shanghai 200031, China; 3NHC Key Laboratory of Myopia, Fudan University, Shanghai 200031, China; 4Shanghai Research Center of Ophthalmology and Optometry, Shanghai 200031, China; 5School of Optometry and Vision Science, University of New South Wales, Sydney, NSW 2052, Australia

**Keywords:** myopia, orthokeratology, corneal astigmatism, lens toricity, elevation difference

## Abstract

This retrospective study investigated the toricity of dual-axis corneal refractive therapy (CRT) orthokeratology lenses and corneal parameters, including flat keratometry (FK), flat eccentricity (*e*), steep *e*, corneal astigmatism, and the difference in elevation at 8 mm chord length. We analyzed the right eyes of 143 adolescent patients who underwent ocular examinations, subjective refraction, and corneal topography before CRT lens fitting by trial lens evaluation. After orthokeratology treatment, all patients underwent a topography map with an intact plus power ring and decentration of <1 mm. The mean patient age was 10.7 ± 2.2 years old; 33% were male. The lens toricity range was 25–100 µm. Multiple linear regression analysis showed significant associations between CRT lens toricity and corneal astigmatism (β = 10.913, t = 3.012, *p* = 0.003) and the difference in elevation at 8 mm chord length (β = 0.681, t = 4.049, *p* < 0.001); no association was found between CRT lens toricity and FK, flat *e*, or steep *e* (all *p* > 0.05). Corneal astigmatism was positively associated with difference in elevation at 8 mm chord length (*r* = 0.743, *p* < 0.001, Pearson’s correlation), and corneal astigmatism and the difference in elevation at 8 mm chord length were positively associated with CRT lens toricity (*r* = 0.657 and *r* = 0.643, respectively; both *p* < 0.01, Spearman’s correlation). These results suggest that difference in elevation at 8 mm chord length can be used to conveniently estimate CRT lens toricity in clinical practice, using the equation Y (CRT lens toricity) = 1.02X (difference in elevation at 8 mm chord length) + 20.3.

## 1. Introduction

Myopia has become a global epidemic, especially in Asian regions, with a prevalence of up to 84% among high school students [1]. In Chinese children, the prevalence of myopia increases exponentially from the age of 7 years to the end of adolescence [2]. Confronted with the growing morbidity and early onset trend of myopia, China faces an ever-growing challenge of myopia control. One strategy that has been shown to be effective for controlling myopia progression in children is orthokeratology (ortho-k) [3,4,5,6,7,8]. This method uses a type of rigid, reverse-geometry-designed, gas-permeable contact lens, which is worn overnight to reshape the cornea and achieve temporary correction of myopia during the day [9]. 

Corneal refractive therapy (CRT) is one of the most frequently used ortho-k lens designs both in the US and China. It utilizes a fitting philosophy of measuring the corneal shape in a sagittal height manner and applies a lens consisting of three zones to achieve the target height of the cornea, yielding a molding effect to the latter. A well-centered lens, indicated by a bullseye topographic pattern after ortho-k treatment, is believed to provide the wearer with optimized visual quality.

Traditionally, ortho-k is contraindicated in patients with high corneal astigmatism, due to poor lens centration and visual outcomes. This severely limits the application of ortho-k in patients with high corneal astigmatism. In response to this limitation, toric or dual-axis ortho-k lenses were developed; they have since demonstrated improved lens centration and more efficacious control of myopia in patients with moderate to high limbus-to-limbus corneal astigmatism [10,11,12,13,14]. However, in the trial lens-fitting of toric ortho-k lenses, the determination of the lens toricity is largely empirical and achieves a reasonable success rate only for experienced practitioners.

This study sought to identify a method for accurately estimating ortho-k lens toricity when fitting an astigmatic cornea. Patients fitted with the CRT dual-axis lens were reviewed retrospectively, and the associations between the final prescribed lens toricity and baseline flat keratometry (FK), flat eccentricity (*e*), steep *e*, corneal astigmatism, and the difference in elevation at the 8 mm chord length (difference in elevation at 8 mm chord length) were investigated.

## 2. Patients and Methods

### 2.1. Patients

This study retrospectively reviewed 143 patients (143 eyes) who presented to the Eye & ENT Hospital of Fudan University and were successfully fitted with dual-axis CRT lenses from 1 February 2018 to 31 August 2019. The inclusion criteria were (1) neophyte contact lens users, (2) baseline spherical refractive error between −4.00 DS to −0.50 DS and cylindrical refractive error not exceeding 1.50 D, (3) uncorrected distance visual acuity no worse than 1.0 (Snellen) after one month of corneal reshaping, and (4) presented with at least two topography maps having a “bullseye” pattern (with lens decentration < 1 mm, classified as Type I and II) during follow-up visits. The exclusion criteria included (1) other ocular diseases, (2) history of ocular surgery, (3) any contraindications to wearing contact lenses, (4) uncorrected distance visual acuity of 0.8 (Snellen) or worse after wearing ortho-k lenses, and (5) lens decentration > 1 mm during ortho-k treatment. 

### 2.2. Refraction

Noncycloplegic manifest refraction was performed to determine the refractive correction target for ortho-k treatment, using spherical equivalent refraction. Cycloplegic autorefraction (ARK-1; NIDEK, Gamagori, Japan) was measured 30 min after cycloplegia with five drops of 0.5% tropicamide eye drops, administered five minutes apart. 

### 2.3. Orthokeratology Lens

This study used dual-axis HDS100 CRT lenses (Paragon Vision Sciences, Gilbert, AZ, USA) with a Dk of 100 × 10^−11^ (cm^2^/s) (mL O_2)_/(mL·mmHg). CRT lenses are composed of an optical zone, reverse zone, and landing zone from the center to the periphery. The total diameter of the prescribed lenses ranged from 10.5 to 12.0 mm based on the patient’s corneal diameter. The back optical zone diameter was 6.0 mm in all the cases. A dual-axis CRT lens has a difference in sagittal depth (in the reverse zone only) along the two principal meridians, ranging from 25 µm to 100 µm, in 25 µm intervals. The purpose of using a dual-axis lens (lens with toricity) in an astigmatic cornea is to better match the corneal shape in order to obtain an equilibrium of hydraulic forces beneath the lens to maximize the possibility of lens centration.

### 2.4. Examinations before Lens Fitting

A routine ocular examination using a slit-lamp biomicroscope was performed for each patient. Other examinations included uncorrected distance visual acuity measurement, autorefraction, and subjective refraction. A Placido ring corneal topographer (Medmont E300; Medmont, Nunawading, Australia) was used to perform corneal topography for each patient. For each measurement, four high-quality maps were collected, repeatability of the four maps was checked, and the one measured with the best tear film quality was selected for analysis. Corneal elevation difference along the two principal meridians at the 8 mm chord length was automatically calculated by the topographer software (Figure 1). The average corneal elevation difference was defined as the elevation difference between the horizontal and vertical meridian.

### 2.5. CRT Lens Fitting

Lens fitting was performed using a CRT diagnostic app/card and trial lens fitting. Each patient was fitted by two experienced doctors (Z.C. and J.Z.). Successful lens fitting was characterized by (1) appropriate centration and adequate movement, (2) a central clearance zone 3–4 mm in diameter, surrounded evenly by a steepened reverse zone, and a parallel landing zone without significant tear leakage underneath the lens during blinking, and (3) adequate edge lift.

### 2.6. Follow-Up

Patients were followed-up at 1 day, 1 week, 1 month, and every 3 months after wearing the lenses. At every follow-up visit, corneal health and uncorrected distance visual acuity were examined, and topography maps were collected.

### 2.7. Classification of Corneal Topography (Difference Map)

After the ortho-k treatment, a difference map was produced by subtracting the pre-ortho-k tangential curvature map from the post-ortho-k tangential curvature map. Eight points on the difference map were plotted surrounding the central flattened area, on which the power was zero. These points were loaded into the MATLAB program to calculate the best-fitting circle using the circle fit function.

The center of the circle was defined as the center of the treatment zone (TxZ), and its distance from the corneal vertex normal was defined as the magnitude of decentration of the ortho-k lens, using methods published previously [15]. The difference maps were classified into the following three types:Type I: intact plus power ring; well centered with decentration ≤ 0.5 mm (Figure 2a).Type II: intact plus power ring; acceptable centration with decentration of ≤1 mm (Figure 2b).Type III: incomplete plus power ring or decentration of >1 mm (Figure 2c).

### 2.8. Statistical Analysis

Statistical analyses were performed using SPSS (Version 24.0, IBM, New York, NY, USA). The Kolmogorov–Smirnov test was used for normality testing. Spearman’s correlation analysis was conducted to analyze the relationships between the lens toricity and FK, flat *e*, steep *e*, corneal astigmatism, and difference in elevation at 8 mm chord length. Pearson’s correlation analysis was conducted to analyze the relationships between the difference in elevation at 8 mm chord length and corneal astigmatism. Multiple linear regression analysis was used to analyze the association between these parameters and lens toricity. Parameters following a normal distribution are expressed as mean ± standard deviation (SD); nonnormal data are presented as the median (interquartile range, IQR). Statistical significance was set at *p* < 0.05.

## 3. Results

### 3.1. Demographics

The mean age of the patients was 10.7 ± 2.2 years (range: 8–19 years); 47 of 143 patients were male (33%). The baseline ocular biometrics and prescribed lens toricities are listed in Table 1; 87 patients had corneal topography type I (61%), and 56 patients had corneal topography type II (39%).

Spearman correlation analysis showed that lens toricity was significantly correlated with both corneal astigmatism (*r* = 0.657, *p* < 0.001; Figure 3a) and the average difference in elevation at 8 mm chord length (*r* = 0.643, *p* < 0.001; Figure 3b). 

The equation that uses corneal astigmatism to predict lens toricity is
Y = 22.75X + 21.75(1)

The equation that uses the average difference in elevation at 8 mm chord length to predict lens toricity is
Y = 1.02X + 20.3(2)

Lens toricity showed no significant correlation with FK, flat *e*, or steep *e* (all *p* > 0.05) (Table 2).

### 3.2. Multiple Linear Regression Analysis

The results showed that lens toricity was significantly associated with corneal astigmatism (β = 10.913, t = 3.012, *p* = 0.003) and the average difference in elevation at 8 mm chord length (β = 0.681, t = 4.049, *p* < 0.001), but not with FK, flat *e*, or steep *e* (all *p* > 0.05) (Table 3).

Pearson’s correlation analysis showed that corneal astigmatism was positively correlated with the average difference in elevation at 8 mm chord length (*r* = 0.743, *p* < 0.001) (Figure 4). The equation using corneal astigmatism to predict the average difference in elevation at 8 mm chord length is
Y = 16.91X + 10.54(3)

## 4. Discussion

This study found that both corneal astigmatism and the average difference in elevation at 8 mm chord length of the cornea were correlated significantly with the toricity of CRT dual-axis lenses. The resultant equation can be used in clinical practice to prescribe a toric-designed ortho-k lens in a straightforward manner, particularly when fitting CRT dual-axis lenses.

Few studies have focused on the relationships between the toricity of an ortho-k lens and corneal parameters. Zhang et al. [16] used artificial intelligence to calculate lens parameters that would match the parameters of the eye, in order to help practitioners make fast and accurate selections of an ortho-k lens. However, that study did not quantify the decentration of the toric lens on corneas, and the developed formula consisted of three components, which made it too complicated for use in clinical practice. Our study categorized the centration by quantitatively measuring the locations of the plus power rings. Among the lenses having good centration, mild decentration, or significant decentration, only the former two were included in the analysis. Thus, only lenses with toricity suitable for astigmatic corneas were evaluated.

Although the toricity of the lens can be predicted with similar reliability using Equations (1) and (2) (*r* = 0.657 and *r* = 0.643, respectively), the latter is more straightforward for clinical use. For example, an average difference in elevation at 8 mm chord length of 30 μm is indicative of a lens toricity of approximately 50 μm. The discrepancy between the toricity of the lens and corneal elevation difference may be the result of the difference between the theoretical and practical landing points of the lens. A CRT lens with a regular back optic zone diameter (6.0 mm) and reverse zone width (1.0 × 2 mm) is very likely to land on the cornea beyond the 8 mm chord length, and the elevation difference will exceed the amount that is within the 8 mm chord length in a typical astigmatic cornea. However, the mean palpebral fissure height value reported in Chinese children is 7.69 ± 1.06 mm [17], making it difficult for practitioners to collect data beyond an 8 mm chord in the vertical meridian. Therefore, the average difference in elevation at 8 mm chord length can be used to estimate the elevation difference at the practical lens-landing point (e.g., at the 9 mm chord), with an average difference of 20 μm.

In the present study, corneal astigmatism was positively correlated with the average difference in elevation at 8 mm chord length (*r* = 0.743, *p* < 0.001). Equation (3) estimates that 1D corneal astigmatism is equivalent to 27.5 μm of difference in elevation at the 8 mm chord length, which is consistent with Tomiyama et al.’s study [18] that found that 1D corneal astigmatism corresponded to a difference of 25 μm in elevation at the 8 mm chord length. Cross-sectional population-based studies have shown that Chinese preschool children have a median of −1.06 D in corneal astigmatism, and 56.1% of them have >1.0 D of corneal astigmatism [19]. Another study investigating Chinese myopic children aged 6–16 years found that the mean corneal astigmatism was 1.12 D [20]. In our clinical setting, 78% of CRT lenses in the past three years had a dual-axis design (unpublished data), which is consistent with the high prevalence of corneal astigmatism in Chinese children and adolescents.

This study’s strength is that we developed a method capable of estimating ortho-k lens toricity using corneal parameters in a straightforward and reliable manner. The equation we developed can help clinicians make quick decisions when fitting CRT ortho-k lenses, saving precious chair time and achieving first-lens centration simultaneously. The first limitation of this study is that the equation can only be used to fit CRT lenses, as ortho-k lenses vary significantly in lens design; therefore, the fitting philosophy cannot be readily extrapolated to other lens designs. The second limitation is that ortho-k, despite its increasing popularity in China, is a relatively small arm of contact lens practice, and necessitates a long learning curve when compared with other types of contact lenses. However, overnight ortho-k has its uniqueness in providing its wearers with good unaided visual acuity during the day and lens centration is the key to success. Therefore, this study is of clinical merit to neophyte ortho-k practitioners.

In conclusion, the toricity of a well-centered dual-axis CRT ortho-k lens is correlated significantly with corneal astigmatism and difference in elevation at 8 mm chord length, but is not correlated with FK, flat *e*, or steep *e*. The equation using the difference in elevation at 8 mm chord length to predict lens toricity is a straightforward method for helping clinicians quickly estimate the parameters for CRT lens fitting.

## Figures and Tables

**Figure 1 jcm-11-05635-f001:**
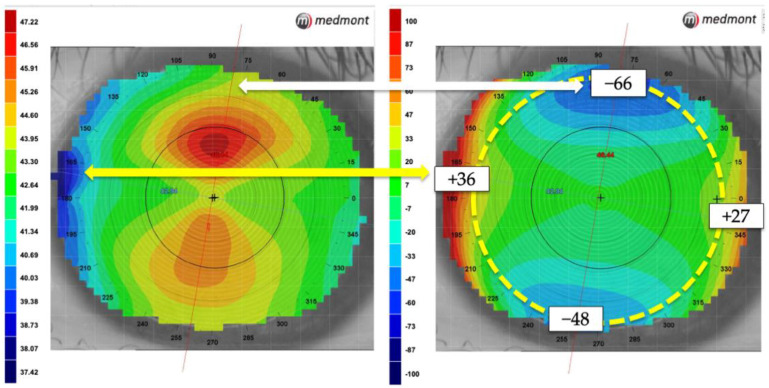
Axial (**left**) and elevation (**right**) map of a with-the-rule astigmatic cornea, with white and yellow arrows showing the correspondence between the two maps of the same corneal locations. In this case, the average elevation difference at the 8mm chord length (yellow dash line) is $\frac{\left(+36)+(+27\right)}{2}-\frac{\left(-66)+(-48\right)}{2}$ = 88.5 µm.

**Figure 2 jcm-11-05635-f002:**
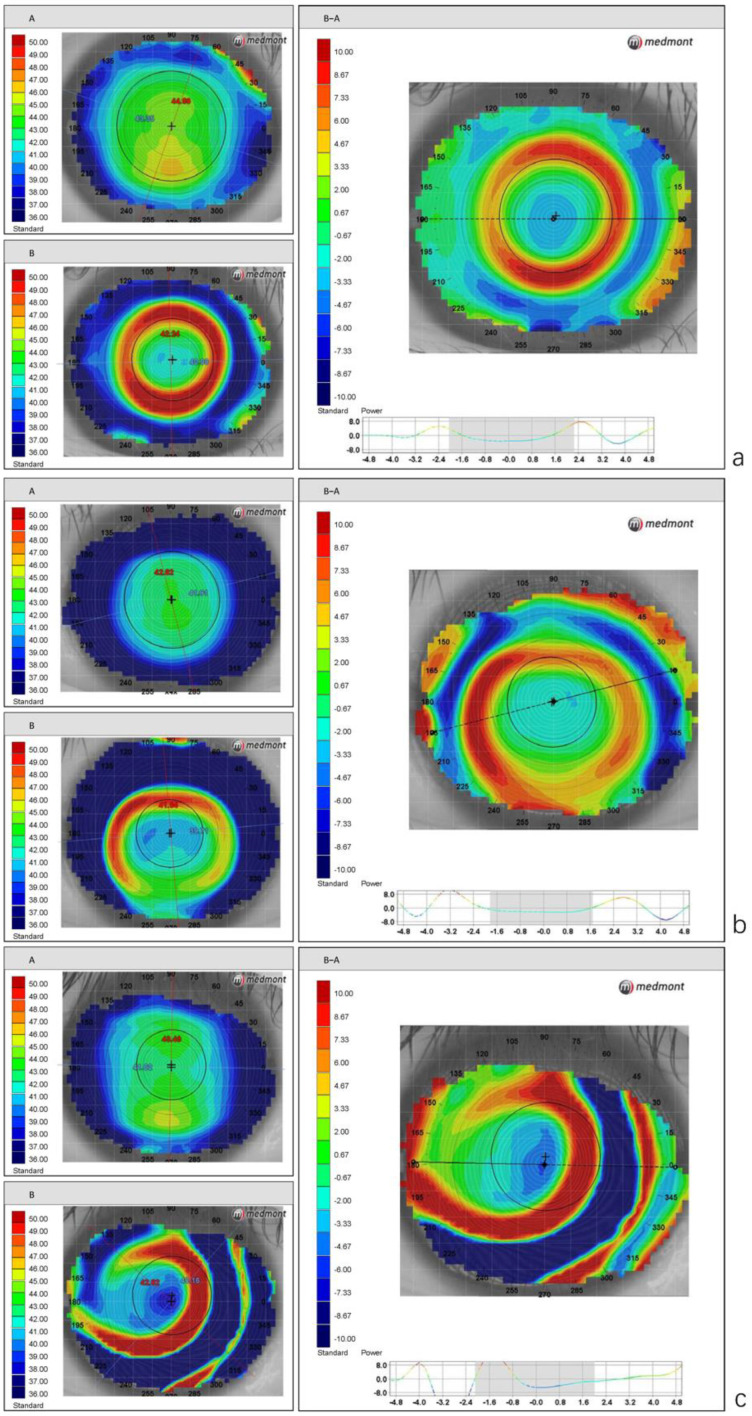
The three types of difference maps after orthokeratology lens wearing, based on the intactness of the plus power ring and amount of decentration. (**a**) Type I: intact plus power ring; well centered with decentration ≤ 0.5 mm. (**b**) Type II: intact plus power ring; acceptable centration with decentration of ≤1 mm. (**c**) Type III: incomplete plus power ring or decentration of >1 mm.

**Figure 3 jcm-11-05635-f003:**
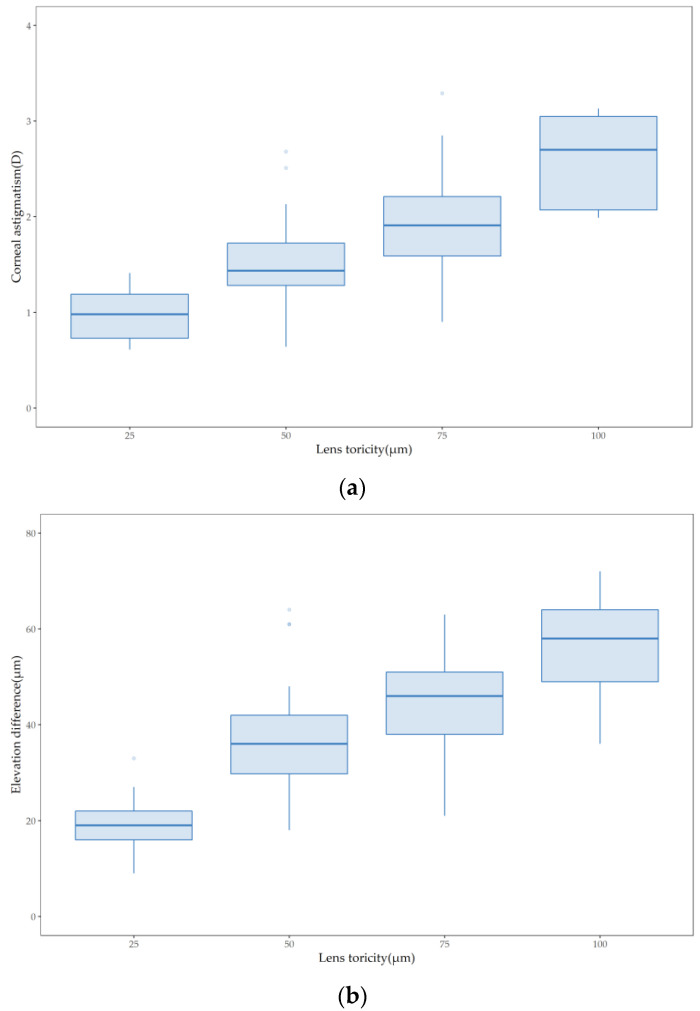
Boxplots showing that a higher lens toricity corresponded to (**a**) a higher corneal astigmatism and (**b**) a higher average elevation difference at 8 mm chord length.

**Figure 4 jcm-11-05635-f004:**
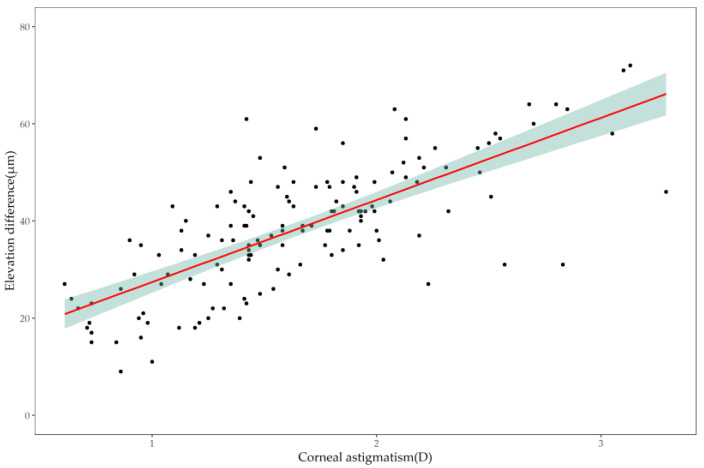
Scatterplot showing the correlation between corneal astigmatism and the average elevation difference at 8 mm chord length of the cornea, with linear trend (red line) and the confidence interval (green shade).

**Table 1 jcm-11-05635-t001:** Baseline ocular biometrics and lens toricity.

Parameter	Minimum	Maximum	Mean/Median	SD/IQR
Spherical degree (D)	−4.00	−0.50	−2.66	0.92
Cylindrical degree (D)	−1.50	0.00	−0.78	0.52
Spherical equivalent (D)	−4.88	−1.00	−3.05	0.97
Flat keratometry (D)	39.11	45.84	42.87	1.24
Flat eccentricity	0.46	0.85	0.65	0.08
Steep eccentricity	0.05	0.82	0.45	0.16
Corneal astigmatism (D)	0.61	3.29	1.66	0.57
Average elevation difference at 8 mm chord length (μm)	9	72	39	13
Lens toricity (μm)	25	100	50	25

Note: The lens toricity data do not follow a normal distribution; therefore, they are expressed as the median and interquartile range (IQR). The remaining parameters follow a normal distribution and are expressed as the mean ± standard deviation (SD).

**Table 2 jcm-11-05635-t002:** Correlation of lens toricity with baseline ocular biometrics using Spearman’s correlation analysis.

Parameter	*r*	*p*
Flat keratometry (D)	0.011	0.893
Flat eccentricity	0.159	0.057
Steep eccentricity	0.074	0.382
Corneal astigmatism (D)	0.657	<0.001
Average elevation difference at 8 mm chord length (μm)	0.643	<0.001

*r*: Spearman’s correlation coefficient.

**Table 3 jcm-11-05635-t003:** Association between toricity of lens and corneal parameters.

Parameter	Unstandardized Beta (β) Coefficients	Standardized Regression Coefficients	*t*	*p*
Flat keratometry (D)	0.157	0.01	0.167	0.868
Flat eccentricity	−2.945	−0.011	−0.141	0.888
Steep eccentricity	7.745	0.063	0.679	0.498
Corneal astigmatism (D)	10.913	0.319	3.012	0.003
Average elevation difference at 8 mm chord length (μm)	0.681	0.453	4.049	<0.001

## Data Availability

The data used to support the findings of this study are available from the corresponding author upon request.

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
