# Peer review of "To Achieve a Bullseye: Factors Related to Corneal Refractive Therapy Orthokeratology Lens Toricity"

_jcm, 2022, doi:10.3390/jcm11195635_

Round 1

Reviewer 1 Report

Because OrthoKeratology lens is a very narrow field of optometry or ophthalmology, I think JCM is not an proper journal for this manuscript. I have several comments about the manuscript.

1. Describle what is the toricity of the lens. Please use some figures to show the characteristics of cornea and toric OK lens. Average elevation difference at 8-mm chord should also be demonstrated.

2. Tile : CRT is not appropriate. Most of the readers don't know the meaning of CRT.

3. The results are not novel. Figure 2 - The results presented are all too natural.

4. Table 1 - Mean/Median? only one value for each variable is presented in the table.

5. I cannot find the clinical significance of this study.

Author Response

Because orthokeratology lens is a very narrow field of optometry or ophthalmology, I think JCM is not an proper journal for this manuscript. I have several comments about the manuscript.

Response: Thank you for your comments. Although ortho-k is a specialty arm of myopia management practice, JCM has published quite a few high-quality papers on ortho-k, some of which are very recent. Here are a few examples:

  • Sánchez-González JM, De-Hita-Cantalejo C, Baustita-Llamas MJ, Sánchez-González MC, Capote-Puente R. The Combined Effect of Low-dose Atropine with Orthokeratology in Pediatric Myopia Control: Review of the Current Treatment Status for Myopia. J Clin Med. 2020;9(8):2371. Published 2020 Jul 24. doi:10.3390/jcm9082371
  • Loertscher M, Backhouse S, Phillips JR. Multifocal Orthokeratology versus Conventional Orthokeratology for Myopia Control: A Paired-Eye Study. J Clin Med. 2021;10(3):447. Published 2021 Jan 24. doi:10.3390/jcm10030447
  • Pauné J, Fonts S, Rodríguez L, Queirós A. The Role of Back Optic Zone Diameter in Myopia Control with Orthokeratology Lenses. J Clin Med. 2021;10(2):336. Published 2021 Jan 18. doi:10.3390/jcm10020336
  • Pereira-da-Mota AF, Costa J, Amorim-de-Sousa A, González-Méijome JM, Queirós A. The Impact of Overnight Orthokeratology on Accommodative Response in Myopic Subjects. J Clin Med. 2020;9(11):3687. Published 2020 Nov 17. doi:10.3390/jcm9113687

Therefore, the article we submitted should meet the JCM special issue “new frontiers in myopia progression in children” and we thank you for your consideration.

Specicific comments:

  1. Describle what is the toricity of the lens. Please use some figures to show the characteristics of cornea and toric OK lens. Average elevation difference at 8-mm chord should also be demonstrated.

Response: Thank you for your comments. The description of lens toricity (Line 83-86) and the definition of corneal elevation difference (Line 93-96 and Figure 1) was added.

  1. Title : CRT is not appropriate. Most of the readers don't know the meaning of CRT.

Response: Thank you for your comments. CRT has been revised as corneal refractive therapy in the title.

  1. The results are not novel. Figure 2 - The results presented are all too natural.

Response: Thank you for your comments. The Figures have been redone for better appreciation of data.

  1. Table 1 - Mean/Median? only one value for each variable is presented in the table.

Response: Thank you for your question. Mean/Median equals mean or median. As put in the note, the lens toricity data does not follow a normal distribution; therefore, it is expressed as the median and interquartile range (IQR). The remaining parameters follow a normal distribution and are expressed as the mean ± standard deviation (SD). (Line 147-149)

  1. I cannot find the clinical significance of this study.

Response: Thank you for your comments. Ortho-k is one of the most important myopia control modalities in China, with over 2 million wearers, most of whom are children. Furthermore, CRT lens is one of the most frequently used lens designs in China. This study, aiming the “new frontiers in myopia progression in children” special issue of JCM, facilites the decision making process for a practitioner on CRT lens parameter selection, and also to obtain a better success rate and in lens fitting. Therefore, we appreciate your consideration for our article to be published on this special issue of JCM.

Reviewer 2 Report

I read with attention the article from Li et al.

They propose a new way of calculating the ortho-K lens toricity, based on its correlation with the average difference in elevation at 8 mm chord length of the cornea.

Anyway, I found few critical points that need to be approached :

The Authors reported that the children enrolled were submitted to slit lamp examination and other exams, but they didn't mention cycloplegic refraction. Were the patients submitted to cycloplegia? What kind of cycloplegic drops were used to assess their refraction?

How did you calculate the amount of myopia to be corrected and where did you prefer to use a spheroequivalent instead of total correction of the astigmatism?

What method of assessment of topographic changes did you adopt? As we are dealing wth rigid gas permeable contact lenses, how many days before the exam did the patients discontinue their contact lens wear?

The Authors revealed that in Chinese children there is a mean palpebral fissure shorter than in other population; how this parameter could influence the results and the significance of the study?

You enrolled in the study patients with low degree of astigmatism, even only 0.50 d…

I think that the main topic needs to be approached with a more complete view, with better evaluation of weaknesses and strengths as to be considered of some impact in the current scenario.

Author Response

I read with attention the article from Li et al.

They propose a new way of calculating the ortho-K lens toricity, based on its correlation with the average difference in elevation at 8 mm chord length of the cornea.

Anyway, I found few critical points that need to be approached :

  1. The Authors reported that the children enrolled were submitted to slit lamp examination and other exams, but they didn't mention cycloplegic refraction. Were the patients submitted to cycloplegia? What kind of cycloplegic drops were used to assess their refraction?

Response: Thank you for your question. Yes the patients underwent cycloplegic autorefraction using 0.5% tropicamide eye drops. The methodology of refraction was added on Line 75-77.

  1. How did you calculate the amount of myopia to be corrected and where did you prefer to use a spheroequivalent instead of total correction of the astigmatism?

Response: Thank you for your question. Spherical equivalent refraction was used as the refractive correction target. Added on Line 74-75.

  1. What method of assessment of topographic changes did you adopt? As we are dealing wth rigid gas permeable contact lenses, how many days before the exam did the patients discontinue their contact lens wear?

Response: Thank you for your question. The pediatric and young adult patients were all neophyte contact lens users, which was added on Line 64. As regard to the methodology of topographic change assessment, we mentioned on Line 114-115: “a difference map was produced by subtracting the pre-ortho-k tangential curvature map from the post-ortho-k tangential curvature map” and on Line 119-121: “The center of the circle was defined as the center of the treatment zone (TxZ), and its distance from the corneal vertex normal was defined as the magnitude of decentration of the ortho-k lens, with the methods published previously”(reference 15)

[15] Chen, Z.; Xue, F.; Zhou, J.; Qu, X.; Zhou, X.; Shanghai Orthokeratology and Study (SOS) Group. Prediction of Orthokeratology Lens Decentration with Corneal Elevation. Optom. Vis. Sci. 2017, 94, 903–907. DOI:10.1097/OPX.0000000000001109.

  1. The Authors revealed that in Chinese children there is a mean palpebral fissure shorter than in other population; how this parameter could influence the results and the significance of the study?

Response: Thank you for your question. Although people of Asian descent are believed to have a tighter lid, which has a potential impact on ortho-k lens centration, there is no evidence leading to this conclusion. Lens centration is largely attributed to the symmetry of the cornea and proper lens design used. (reference 15)

[15] Chen, Z.; Xue, F.; Zhou, J.; Qu, X.; Zhou, X.; Shanghai Orthokeratology and Study (SOS) Group. Prediction of Orthokeratology Lens Decentration with Corneal Elevation. Optom. Vis. Sci. 2017, 94, 903–907. DOI:10.1097/OPX.0000000000001109.

  1. You enrolled in the study patients with low degree of astigmatism, even only 0.50 d…

Response: Thank you for your comments. On one hand, although corneal elevation is significantly positively correlated with corneal astigmatism (Figure 4), the scatterplot showed remarkable individual variation among the patients. For example, the ones with ≈1D of corneal astigmatism revealed an elevation difference ranging from 10 μm to 45 μm. Therefore, ≈0.5D of corneal astigmatism does not necessarily mean that a dual-axis lens is not needed. On the other hand, adopting a wide range of corneal elevation along with corneal astigmatism granted us a greater chance to address the research question as which parameter is more reliable in deciding the toricity of the lens. The results indicated that the parameters are predictive of the lens toricity with very similar reliability.

  1. I think that the main topic needs to be approached with a more complete view, with better evaluation of weaknesses and strengths as to be considered of some impact in the current scenario.

Response: Thank you for your suggestion. We added some contents to the manuscript in order to make the article more friendly to general readers.

  • Title: To achieve a bull’s-eye: factors related to corneal refractive therapy orthokeratology lens toricity
  • Introduction (Line 43-47): Corneal refractive therapy (CRT) is one of the most frequently used ortho-k lens designs both in the US and China. It utilizes a fitting philosophy of measuing the corneal shape in a sagittal height manner and applies a lens consisting of three zones to achieve the target height of the cornea, yielding a molding effect to the latter. A well-centered lens, indicated by a bull’s-eye topographic pattern after ortho-k treatment, is believed to provide the wearer with optimized visual quality.
  • Discussion (Linx 219-229): This study’s strength is that we developed a method capable of estimating ortho-k lens toricity using corneal parameters in a straightforward and reliable manner. The equation we developed can help clinicians make quick decisions when fitting CRT ortho-k lenses, saving precious chair time and achieving first-lens centration simultaneously. The first limitation of this study is that the equation can only be used to fit CRT lenses, as ortho-k lenses vary significantly in lens design; therefore, the fitting philosophy cannot be readily extrapolated to other lens designs. The second limitation is that ortho-k, despite its increasing popularity in China, is a relatively small arm of contact lens practice, and necessitates a long learning curve when compared with other types of contact lenses. However, overnight ortho-k has its uniqueness in providing its wearers with good unaided visual acuity during the day and lens centration is the key to success. Therefore, this study is of clinical merit to neophyte ortho-k practitioners.

Round 2

Reviewer 1 Report

The manuscipt has been improved.

I feel that my concerns are resolved.

Author Response

Thank you so much for reviewing the manuscript and your comments to improve it. Since the reviewer's concerns are resolved, we look forward to the proof from the publishing house soon.

Kind regards